# AM3F-FlowNet: Attention-Based Multi-Scale Multi-Branch Flow Network

**DOI:** 10.3390/e25071064

**Published:** 2023-07-14

**Authors:** Chenghao Fu, Wenzhong Yang, Danny Chen, Fuyuan Wei

**Affiliations:** 1School of Information Science and Engineering, Xinjiang University, Urumqi 830017, China; 107552103626@stu.xju.edu.cn (C.F.); kabuoxygen@163.com (D.C.); wfy@stu.xju.edu.cn (F.W.); 2Xinjiang Key Laboratory of Multilingual Information Technology, Xinjiang University, Urumqi 830017, China

**Keywords:** micro-expression recognition, attention mechanisms, logit-adjusted loss

## Abstract

Micro-expressions are the small, brief facial expression changes that humans momentarily show during emotional experiences, and their data annotation is complicated, which leads to the scarcity of micro-expression data. To extract salient and distinguishing features from a limited dataset, we propose an attention-based multi-scale, multi-modal, multi-branch flow network to thoroughly learn the motion information of micro-expressions by exploiting the attention mechanism and the complementary properties between different optical flow information. First, we extract optical flow information (horizontal optical flow, vertical optical flow, and optical strain) based on the onset and apex frames of micro-expression videos, and each branch learns one kind of optical flow information separately. Second, we propose a multi-scale fusion module to extract more prosperous and more stable feature expressions using spatial attention to focus on locally important information at each scale. Then, we design a multi-optical flow feature reweighting module to adaptively select features for each optical flow separately by channel attention. Finally, to better integrate the information of the three branches and to alleviate the problem of uneven distribution of micro-expression samples, we introduce a logarithmically adjusted prior knowledge weighting loss. This loss function weights the prediction scores of samples from different categories to mitigate the negative impact of category imbalance during the classification process. The effectiveness of the proposed model is demonstrated through extensive experiments and feature visualization on three benchmark datasets (CASMEII, SAMM, and SMIC), and its performance is comparable to that of state-of-the-art methods.

## 1. Introduction

Micro-expressions are usually unconscious and respond to stimuli (e.g., an emotional event or stressful situation) [1]. Due to their unconscious nature, micro-expressions are considered more reliable indicators of emotion than conscious facial expressions. As a result, micro-expression recognition has received increasing attention in various fields, such as psychology, criminology, and human–computer interaction.

Micro-expression recognition aims to classify micro-expression videos into different emotional categories. In each micro-expression video, the frame where the micro-expression face action starts is noted as the start frame, while the end frame is the offset frame, and the frame with the highest intensity is the vertex frame [2]. Like facial expression recognition, micro-expression recognition classifies facial images/sequences into sadness, surprise, happiness, etc. However, interpreting micro-expressions is more challenging for humans, and micro-expression recognition studies perform poorly. There are two reasons for the low performance of micro-expression recognition research: (1) the lack of micro-expression datasets, which are often required for complex deep learning systems [3]; and (2) the low intensity of micro-expressions makes it challenging to extract salient and distinguishing features. However, building new micro-expression datasets is a time-consuming and challenging task. Several existing studies have found that significant progress can be made in automatic micro-expression analysis by designing effective shallow networks and exploring attention mechanisms. Deep-learning-based micro-expression recognition methods have achieved state-of-the-art performance. Therefore, the optimal solution is to process the micro-expression sample into a distinct feature and then use deep learning techniques for the classification task.

Due to a limited amount of data, single-scale features may not be sufficient to distinguish different micro-expression categories. Combining multiple channels and effectively integrating various scale features can improve the ability of the model to learn micro-expression features. Therefore, we propose an attention-based multi-scale, multi-branch flow network for mining of subtle micro-expression movements from a limited micro-expression dataset. Specifically, in the preprocessing data stage, we compute the optical flow using the initial and vertex frames of the micro-expression video and generate new optical flow information (optical strain) [4] to better capture the subtle facial motion of micro-expressions. Second, we use the inception network [5] to input the three pieces of optical flow information into the three-branch network for feature extraction. Considering that single-scale features cannot adequately extract local motion, we propose a new multi-scale feature fusion (MSFF) module that focuses on the critical information at each scale using a spatial attention mechanism. Then, since each piece of optical flow information contributes differently to micro-expression classification, we propose a multi-optical flow feature reweighting (MOFRW) module that adaptively selects features for each optical flow separately using the channel attention mechanism [6]. Finally, we introduce a logit adjustment loss [7] to weight the prediction scores of micro-expression categories using prior knowledge to effectively balance unevenly distributed samples to further improve the performance of our model. The contributions of this paper can be summarized as follows:In the AM3F-FlowNet framework, we designed a module called MOFRW. This module uses a channel attention mechanism to first score and weight the contribution of each piece of optical flow information, then adaptively select features for the channels within each optical flow feature. This double-weighting approach is effective in highlighting key features and suppressing redundant features.We propose the MSFF module, which enables AM3F-FlowNet to learn the local detail of micro-expression facial movements by fusing information from different scales. The experimental results show that combining different scales to use the underlying stack information entirely plays a crucial role in recognizing locally fine micro-expression movements.Considering the category imbalance in the micro-expression dataset, we introduce a logit adjustment loss function that uses prior knowledge to weight the prediction scores of minority and majority class samples to mitigate the negative impact of category imbalance that may occur during the classification process.Our proposed method is evaluated not only on multiple micro-expression datasets but also on a composite dataset formed by combining multiple micro-expression datasets. The experimental results show that our method achieves intensely competitive performance with state-of-the-art methods.

This paper is organized as follows. In Section 2, we review the work related to this study. In Section 3, we describe our proposed algorithm in detail. Section 4 reports the experimental results of the composite dataset benchmark and the evaluation of a single dataset with an ablation study and visualization analysis of our proposed module. In Section 5, we summarize our work.

## 2. Related Work

### 2.1. Micro-Expression Recognition

As research on micro-expression recognition has intensified in the past few years, more methods have emerged. These methods can be broadly classified into two categories: traditional methods based on conventional machine learning and those based on deep learning.

#### 2.1.1. Traditional Machine Learning

In the early stages of micro-expression research, researchers mainly used image processing and computer vision techniques, such as local binary patterns, gradient operators, and optical flow, to extract hand-designed features from micro-expression videos and traditional machine learning algorithms to classify these features. For example, Yan et al. [8] extracted LBP-TOP features from a spatiotemporal perspective to describe micro-expressions and used SVM classifier for recognition. Wang et al. [9] used LBP-Six Intersection Point (LBP-SIP) to reduce redundant features and provide a more compact and lightweight representation. Li et al. [10] investigated LBP-TOP, directed gradient histogram, and image gradient direction histogram as three feature descriptors and combined them to obtain combined feature vectors for recognition. In addition, some studies used optical flow features based on interframe luminance intensity variations to estimate frame-level motion. Shreve et al. [11] used optical strain, i.e., the optical flow derivative, as a feature descriptor and achieved good results in both macro- and micro-expression recognition. Liu et al. [12] proposed main directional mean optical flow (MDMO) to calculate facial motion, effectively reducing the computational cost by dimensionalizing the features. Liong et al. [13] proposed bidirectional weighted oriented optical flow (Bi-WOOF), which uses only the optical flow information of the start frame and vertex frame to represent the motion changes of the whole micro-expression video, reducing a large number of redundant features. However, these methods are sensitive to data quality and noise, lack robustness, and often face performance bottlenecks.

#### 2.1.2. Deep Learning

With the development and application of deep learning in the field of image recognition, more and more researchers have started to explore deep-learning-based micro-expression recognition methods to overcome the limitations of traditional methods. Patel et al. [14] used a convolutional neural network trained on a macro-expression database as a feature extractor for micro-expression classification using a transfer learning technique—the first application of deep learning to micro-expression recognition. However, the extractor did not outperform some traditional manual methods because micro-expression data still have the problem of small samples, and complex network models tend to lead to overfitting. Quang et al. [15] proposed a capsule network for micro-expression recognition using only apex frames to find part–whole relationships. This method outperformed the provided LBP-TOP baseline method. However, extracting salient and distinguishing features using only raw images/sequences is difficult. Gan et al. [16] combined manual features and deep learning to extract optical stream features from each video’s onset and apex frames, then input the horizontal and vertical components of the optical stream into a dual-stream CNN to learn and extract features automatically using a neural network. Liong et al. [17] proposed a shallow triple-stream 3D CNN, which combines horizontal optical flow, vertical optical flow, and optical strain to learn compact and discriminative feature representations. Zhou et al. [18] designed a dual inception network using horizontal and vertical optical flow features extracted from the onset and apex frames to extract rich information through multiple convolutional kernels of different sizes in parallel. These methods achieved the best results and demonstrated the superiority of combining optical flow with shallow neural networks.

### 2.2. Attention Mechanism in Micro-Expression Recognition

The process of human perceptual analysis is based on attentional mechanisms. Human vision acquires target areas in a scene that require focused attention and focuses more attention on the target area to obtain more detailed information about the scene and thus better capture the visual structure of the target [19]. In computer vision, attention mechanisms are used to discover an image’s interest intervals and highlight the representation of the intervals of interest. The two main aspects of the attention mechanism in deep learning are (1) adaptively acquiring meaningful channels for input features and (2) selecting focus locations to produce more discriminative feature representations.

In micro-expression recognition tasks, attention mechanisms guide the model to focus on critical facial regions and suppress irrelevant facial regions and noise [20]. Researchers have proposed various attentional mechanisms for micro-expression recognition. For example, Wang et al. [21] introduced a micro-attention mechanism in concert with a residual network. This mechanism utilizes self-learning multi-scale features in the residual network architecture to compute an attention map that enables the network to focus on the regions of facial interest and cover different action units to focus on the regions where micro-expressions occur. Yang et al. [22] proposed a convolutional neural network based on the attention mechanism. They embedded generic attention into static face key points, dynamic information into dynamic attention, and semantic attributes related to expressions into channel attention. These three attention mechanisms are integrated to obtain a more differentiated visual representation. Li et al. [23] proposed a muscle-motion-guided network to model local subtle muscle motion patterns by introducing successive attention blocks and introducing the attention map of the previous layer as prior knowledge to generate the attention map of the current layer. Su et al. [24] proposed a micro-expression extractor guided by a critical face component recognition method that uses a face semantic segmentation probability map involving multiple essential facial parts to guide face feature learning and highlights the relevant regions through a component-aware attention module. Zhang et al. [25] proposed an attention-enhanced residual block that uses a triple attention mechanism in the residual block to extract relatively coarse-grained features using channel and spatial attention and multi-headed self-attention on different spatial locations and channels of the features to generate a self-attention map, which makes the network focus more on critical facial regions.

We propose an MSFF module and an MOFRW module to focus on salient features of facial regions from channel and spatial perspectives based on the attention mechanism. This operation improves the network’s generalization ability and enhances the recognition of different micro-expression types. With these improvements, our model achieves better performance in micro-expression recognition tasks.

## 3. Proposed Method

In this section, we describe the framework of the micro-expression recognition task in detail. The overall flow is shown in Figure 1.

### 3.1. Data Preprocessing

#### 3.1.1. Face Cropping and Apex Frame Positioning

Among the three generic datasets (CASMEII [8], SAMM [26], and SMIC [27]), the CASME II and SMIC datasets provide video sequences of already cropped face images. Since the SAMM dataset does not provide cropped images, we use the Dlib toolkit to detect critical points on the face and crop the images according to these points. All cropped images are resized to a size of 170 × 140.

The subtlety of micro-expression facial muscle movements makes facial changes between two successive frames less obvious. Inspired by [13], the apex frame with the highest motion intensity can provide a meaningful enough representation to encode facial micro-expressions. Therefore, each micro-expression sample can be represented using only the onset and apex frames. For the SMIC dataset with only the onset and offset frames labeled, we use an automatic apex localization algorithm (i.e., D&C-RoIs) [13] to locate the apex frames of each micro-expression sequence. The method first divides the face image into three facial subregions, i.e., mouth, left eye and left eyebrow, and right eye and right eyebrow. Then, LBP features are calculated for each subregion, and the differences in LBP features between the three ROIs are compared with the first frame. The ROI with the most significant difference is selected. The frame with the most considerable intensity of facial muscle movement is searched using a partitioning strategy to obtain the index of micro-expression apex frames. Using this method, D&C-ROIS can efficiently locate micro-expression apex frames and improve the accuracy of micro-expression recognition.

#### 3.1.2. Optical Flow Feature Extraction

The optical flow method is a commonly used method to study dynamic objects and is also applicable to describe the minute movements of facial muscles [28]. Because micro-expression datasets are collected under strict lighting conditions, the light-sensitive nature of the optical flow method can be mitigated to some extent [29]. Compared with texture features, the optical flow method can effectively reduce the domain difference between different databases, which is crucial in improving cross-database micro-expression recognition performance [30]. Therefore, the optical flow method has essential applications in micro-expression analysis.

We use the total variation-l1 optical flow (TV-L1) method to calculate the optical flow between the start and vertex frames. This algorithm is suitable for motion analysis, where the displacement of two adjacent frames is small and the edge feature information of the image can be preserved. We use *u* and *v* to denote the horizontal and vertical components of the optical flow field, respectively, to describe the motion information between the start frame and the vertex frame. The authors of [17] further extracted optical strain from the optical flow to emphasize the fine motion on a tiny scale. Inspired by this, we calculate the optical strain (os) as the third optical flow information according to *u*, *v*:(1)$$os=\frac{1}{2}\left[\nabla O_{f}+{\left(\nabla O_{f}\right)}^{T}\right]$$
where $O_{f}$ = ${\left[u,v\right]}^{T}$ is the optical flow vector, including horizontal and vertical components, and ∇ is the derivative of $O_{f}$. In summary, the following three optical flows can represent each micro-expression sample.

*u*: the horizontal component of the optical flow field;*v*: the vertical component of the optical flow field;os: optical strain.

### 3.2. Network Architecture

#### 3.2.1. Backbone Selection

Larger convolutional kernels usually have larger perceptual fields for convolutional neural network models. They can better capture the global features in the input data, while smaller convolutional kernels are more suitable for extracting local features. Due to the differences in the location of critical information in different optical flow images, choosing the appropriate convolution kernel size for convolutional operations is more complicated. Inspired by [18], we adopt the inception technique, using multiple size filters in parallel at the same level. The inception network extracts features at different scales and abstraction levels, which can be better adapted to graphs of different sizes and complexities, enabling the network to capture richer information. In addition, inception blocks use small convolutional layers of 1 × 1, 3 × 3, and 5 × 5 instead of large filters, while an additional 1 × 1 convolutional layer is added to limit the number of input channels. This design can effectively reduce the number of model parameters and computational effort and improve computational efficiency. Finally, using the maximum pooling operation after each inception block can further reduce the size and computation of the feature maps and filter out the noise and redundant information in the input feature maps to better aggregate the feature information. The inception structure used in this paper is shown in Figure 2.

#### 3.2.2. MFSS Module

The MSFF module proposed in this paper aims to obtain motion information of face optical flow images at different scales. To enhance the salience and expressiveness of features in each layer, we first employ a spatial attention module (SAM) for adaptive weighting, as shown in Figure 3a. The SAM module can learn the features of the previous and current layers, generate the corresponding attention mapping (Attn), and multiply it by the input features to enhance the expression of local information of the face and the representation of motion region features.

Our proposed MSFF module employs a three-step strategy to capture more local motion information in low-level optical flow features, as shown in Figure 3b. (i) To connect with the current layer features, the features of the previous layer are downsampled. Then, the downsampled features are subjected to SAM processing, and before entering SAM, the features are subjected to a summing operation with the enhanced features. (ii) SAM processing is conducted for the current layer. (iii) The processed features of the previous and current layers are concatenated to obtain the final multi-scale fused features (OF-MSFF) of the middle and low neighbor layers. This design can extract and fuse different motion information at different levels and scales to enhance the expression and recognition accuracy of local information on the face.

#### 3.2.3. MOFRW Module

The MOFRW module proposed in this paper stitches the optical flow information of different modes. It uses the channel attention module (CAM) to weigh each piece of optical flow information, as shown in Figure 4a. This design can fully use the characteristics and advantages of different modal optical flow information to improve the performance and differentiation of optical flow features and thus obtain more accurate and reliable dynamic information. When performing CAM and weighting calculation, it is also necessary to weight each channel according to its importance to strengthen critical information and further suppress noise and redundant information to improve the performance and differentiation ability of optical flow features.

The MOFRW module achieves attention weighting of multiple pieces of optical flow feature information through four main steps. The overall process is shown in Figure 4b. First, the three optical flow features are stitched together. The channel attention vectors of different optical flow information feature maps are extracted using CAM to obtain the channel attention of the three optical flow features. Second, the three optical flow channel attention vectors are feature-recalibrated using softmax to obtain new attention weights for each optical flow information interaction. Then, a dot-product operation is performed on the recalibrated weights and the corresponding feature maps. After attention weighting the multi-optical flow feature information, the output is obtained as a feature map. Finally, the three input-spliced optical flow features are summed with the attention-weighted features and output to obtain a reweighted multi-optical flow-weighted feature map. The computational process of the MOFRW module can be described as follows.
(2)$$\begin{array}{cc} Z_{i} & =CAM(F_{i}),i=u,v,os\\ att_{i} & =Softmax\left(Z_{i}\right)=\frac{exp(Z_{i})}{\sum_{i=1}^{3}exp\left(Z_{i}\right)},i=u,v,os\\ Y_{i} & =F_{i}⊙att_{i},i=u,v,os\\ Out & =F_{input}+Cat\left(\left[Y_{u},Y_{v},Y_{os}\right]\right) \end{array}$$

### 3.3. Logit-Adjusted a Priori Weighted Loss

Due to the difficulty of collecting micro-expressions and the significant difference in the difficulty of collecting different types of micro-expressions, the micro-expression dataset has the problem of a highly uneven distribution of categories. To solve this problem, we introduce the logit-adjusted softmax cross-entropy (LASCE) method to balance the prediction scores of minority- and majority-class samples. In the training process, the LASCE method weights the prediction scores of minority and majority samples based on the sample label frequency, which avoids the situation in which majority samples suppress minority samples and optimizes the model training process. Especially in the case of highly unbalanced data distribution, the learning classifier based on the LASCE method can reduce the probability of misclassifying minority class samples as the majority class, thus improving the robustness of the model. In LASCE, the category prior probabilities are summed with the model’s output, then trained using a cross-entropy loss function. Specifically, LASCE is formulated as follows. First, the category prior probability ($P_{p}$) is calculated, where $P_{y}$ denotes the frequency of occurrence of each category, n is the total number of samples, $y_{i}$ is the category of the ith sample, *C* is the total number of categories, and $\left[y_{i}=y\right]$ denotes 1 when $y_{i}=y$ and 0 otherwise:(3)$$p_{y}=\frac{\sum_{i=1}^{n}\left[y_{i}=y\right]}{n},y=1,2,\ldots ,C$$
Then, the correction factor ($a_{y}$) is calculated for each category based on the category prior probability ($P_{p}$), where τ is a hyperparameter and ϵ is a minimal value to avoid the correction factor becoming infinite:(4)$$a_{y}=\log\left(p_{y}^{\tau }+\epsilon \right),y=1,2,\ldots ,C$$
Eventually, the vector ($P_{p}$) consisting of the correction factor ($a_{y}$) is summed with the output of the model and trained using the cross-entropy loss function, i.e.,
(5)$$L_{LASCE}=-\frac{1}{N}\sum_{i=1}^{N}\sum_{j=1}^{C}\left(p_{ij}+a_{j}\right)\log\left(\frac{\exp(p_{ij}+a_{j})}{\sum_{k=1}^{C}\exp\left(p_{ik}+a_{k}\right)}\right)$$
where *N* is the total number of samples, *C* is the total number of categories, $p_{ij}$ is the original predicted value of the *i*th sample belonging to the *j*th category, log denotes the natural logarithm, and exp denotes the natural exponential function. When τ = 0, LASCE is equivalent to the ordinary softmax cross entropy. In contrast, when the value of τ increases, the model training process focuses more on a few classes of samples, and after extensive experiments, we select τ = 0.5.

## 4. Experiments

### 4.1. Datasets

This section aims to validate the feasibility and effectiveness of our proposed emotion recognition method on different datasets. We tested on the CASME II, SAMM, and SMIC datasets. These datasets contain rich emotional information and micro-expression videos, which are highly challenging for emotion recognition tasks.

The CASME II dataset contains 256 micro-expression videos from 26 participants with a resolution of 640 × 480 and a frame rate of 200 FPS. The coder categorizes each video into seven emotion categories: disgust, happiness, depression, surprise, sadness, fear, and other. The SAMM dataset contains 159 micro-expression videos from 32 participants with a resolution of. Each video is categorized by the coder into seven affective categories based on FACS: anger, happiness, surprise, contempt, disgust, fear, sadness, and other. The SMIC dataset contains 164 micro-expression videos from 16 participants with a resolution of 640 × 480 and a frame rate of 100 FPS. These videos are categorized into positive, negative, and surprise emotional categories. We did not consider the categories with smaller sample sizes for the individually tested datasets. Thus, the CASME II, SAMM, and SMIC-HS datasets have 246 (5 categories), 136 (5 categories), and 164 (3 categories) samples, respectively.

In addition to the three datasets mentioned above, we also use the combined dataset proposed by MEG2019 to validate the performance of our algorithm. This dataset labels samples from the CASME II, SAMM, and SMIC datasets into three affective categories: positive, negative, and surprised. We used the same classification strategy as the mainstream approach and tested it on the combined dataset. Table 1 shows the relevant information for the combined dataset.

### 4.2. Evaluation Metrics

In our experiments, we used F1 score (F1) and accuracy (Acc) as evaluation metrics to assess the performance of our sentiment recognition algorithm against the state-of-the-art methods for the problem of uneven class distribution of micro-expression datasets. Specifically, F1 is the summed average of precision and recall, which is used to measure the accuracy and robustness of the classifier. Furthermore, Acc is the ratio of the number of samples correctly classified by the classifier to the total number of samples, which is used to measure the overall accuracy of the classifier. Our evaluation metrics, F1 and Acc, were chosen because they eliminate the bias caused by the uneven distribution of categories in the micro-expression dataset while enabling an objective assessment of the algorithm’s performance. Our comparison with state-of-the-art methods aims to verify whether our algorithm performs better and more robustly.
(6)$$Acc_{c}=\frac{TP_{c}}{N_{c}}$$
(7)$$F1_{c}=\frac{TP_{c}}{2TP_{c}+FP_{c}+FN_{c}}$$
where $F1_{c}$ is the F1 score of the classification results of class *c*; $TP_{c}$, $FP_{c}$, and $FN_{c}$ are the numbers of true positives, false positives, and false negatives in the classification results of class *c*, respectively; $N_{c}$ represents the number of samples in class *c*; and $Acc_{c}$ is the accuracy rate of class *c*, i.e., the proportion of correct predictions in class *c* to the total number of samples in that class.

We used the unweighted F1 score (UF1) and the unweighted average recall (UAR) as evaluation metrics on the combined dataset, aiming to objectively assess the performance and effectiveness of our proposed sentiment recognition algorithm. The reason for using this metric is that the combined dataset contains multiple datasets, with different sample sizes and category distributions in each dataset, thus requiring an unweighted metric to eliminate the differences between the individual datasets. Specifically, UF1 is the average of the F1 score of each category, and UAR is the average of the recall of each category. They can be expressed as:(8)$$UF1=\frac{1}{C}\sum_{c}F1_{c}$$
(9)$$UAR=\frac{1}{C}\sum_{c}Acc_{c}$$
where *C* is the number of micro-expression tags.

### 4.3. Experimental Setup

In this study, we combine three pieces of optical flow information (horizontal optical flow, vertical optical flow, and optical strain) into a 28 × 28 image using the horizontal flip and color dithering techniques for data enhancement. Before the images enter the network, we use the channel dimension to slice the three pieces of optical flow information so that they can enter the multi-branch network. To evaluate the experimental results for the triple and quintuple classification tasks, we use the leave-one-subject-out (LOSO) cross-validation method. Specifically, we use each subject group as the test set and all the remaining subject groups for training. This cross-validation approach can effectively reduce overfitting and improve the robustness and reliability of the model. We used a model based on the PyTorch framework implementation in our experiments and performed all experiments on a TITAN RTX3080. We used the Adam optimizer and set the weight_decay parameter to 1 × 10${}^{-4}$. We set the batch size to 8, the initial learning rate to 0.0001, and the epoch value to 70. These parameters were chosen based on our experience in the experiments and the experimental results to adjust them.

### 4.4. Results and Analysis

In this study, experiments were conducted on the widely used SMIC-HS, CASME II, and SAMM micro-expression databases, as well as the MEGC2019 combined database, and compared with traditional machine learning methods and more recent and prominent deep learning methods to evaluate the performance of the proposed approach. The experimental results are shown in Table 2 and Table 3.

Overall, the method proposed in this paper performed well in the task of micro-expression recognition. In a single-dataset experiment, the proposed method achieved state-of-the-art results on the CASME II database, with an increase in ACC and F1 of 8.714% and 0.0758, respectively, compared to the second-best method. However, compared with the CASME II dataset, this algorithm performed worse on the SAMM and SMIC datasets. On the SMIC-HS dataset, the ACC of the proposed method was 74.25%, representing an improvement of 1.08% over the second-best method, while the F1 was 0.7254, which is slightly lower than the best result. CASME II samples were captured at high frame rates and provided more accurate vertex frames, resulting in more precise optical flow calculations and better depiction of motion changes. In contrast, on the SMIC dataset, the additional vertex frame-locating operation may introduce potential errors. Additionally, SMIC videos were recorded at a lower frame rate (100 fps) and were influenced by various background noises, such as shadows, highlights, and flickering lights caused by the database stimulation settings. For the SAMM dataset, the performance of the proposed algorithm was not ideal due to the smaller sample size for the five-class task, which was just slightly over half of the sample size of the CASME II dataset, and the more severe class imbalance. This once again confirms that the impact of small sample sizes and imbalanced class distributions on classification results cannot be ignored. From the perspective of different emotion categories, the more sample data are available, the more validated evidence for improved recognition rates of specific micro-expression types.

This study also achieved excellent results on a comprehensive dataset. For comparison with the state-of-the-art BDCNN method, we conducted five experiments using the AM3F-FlowNet model on the composite dataset and calculated the average results. According to the experimental results, we achieved state-of-the-art results on the FULL, CASMEII, and SMIC-HS datasets, while on the SAMM dataset, our proposed method slightly fell behind BDCNN but still outperformed other methods. These results demonstrate the effectiveness and generalization ability of the proposed method.

### 4.5. Ablation Experiment

Through the above experiments, we demonstrated the superior performance of AM3F-FlowNet relative to state-of-the-art micro-expression recognition methods. We believe that the improved performance of AM3F-FlowNet can be attributed to its three key components: the MSFF block, MOFRW block, and LASCE. To investigate their effectiveness, we performed ablation experiments on them.

(1) Effectiveness of the MSFF block: To verify the validity of the MSFF block, we compared three cases: (i) no multi-scale is used in AM3F-FlowNet; (ii) no MSFF block is used in AM3F-FlowNet; and (iii) AM3F-FlowNet. According to the results shown in Table 4, AM3F-FlowNet with MSFF blocks outperforms its competitors. For example, with multi-scale, the Acc and F1 of the model improved by 2.09% and 0.0326, respectively, over the prototype, while with an MSFF block on top of multi-scale, the Acc and F1 of the model improved by 3.35% and 0.0383, respectively. The above results demonstrate the key role of using multi-scale to exploit the underlying stack information for micro-expression recognition performance and validate the role of the MSFF block in the effectiveness of AM3F-FlowNet.

(2) Effectiveness of MOFRW block: To verify the effectiveness of the MOFRW block, we compared two cases: (i) mo MOFRW block is used in AM3F-FlowNet; and (ii) AM3F-FlowNet. According to the results shown in Table 4, the MOFRW block has a significant role in improving the model performance by adaptively weighting the features by measuring the contribution of each optical flow to the recognition results. The AM3F-FlowNet with an MOFRW block improved Acc and F1 by 2.93% and 0.0196, respectively, relative to the prototype, validating the effectiveness of MOFRW.

(3) Validity of LASCE: To verify the validity of LASCE, we compared four cases: (i) standard cross-entropy loss function; (ii) focal loss function; (iii) logit-adjusted focal loss function (LA-Focal); and (iv) LASCE. According to the results shown in Table 5, LASCE achieves the best performance in micro-expression recognition, with Acc and F1 improving by 2.12% and 0.0238, respectively, over the standard cross entropy. The underperforming focal loss function also improves after logarithmic adjustment based on prior knowledge. To validate the ability of LASCE to alleviate category imbalance, we present the confusion matrices of the loss function classification results before and after logarithmic adjustment, as shown in Figure 5. Among the five affective categories considered, the accuracy of disgust, happiness, and repression, on the other hand, were increased by employing log-adjusted standard cross-entropy loss and focus loss, resulting in a more balanced distribution of prediction accuracy. Although this was achieved by slightly reducing the accuracy of the major classes, the average accuracy of these three classes exceeded the accuracy of the unweighted loss function. This is especially important when the distribution of classes is extremely unbalanced or when the accuracy of minor classes is more important than that of major classes. These experimental results validate the effectiveness of LASCE in AM3F-FlowNet.

### 4.6. Feature Visualization

To improve the interpretability of models, we usually need to use visualization methods to present the model prediction results. In this process, the heat map is a standard visualization tool that can help us better understand the prediction results of the model. In particular, heat maps can be helpful when verifying the validity of the model’s recognition of specific emotions and determining whether height activation locations correspond to specific motion locations, as shown in Figure 6.

In our experiment, we selected two sets of combinations of optical flow and optical variation images from five emotion categories as our original images. We returned heat maps using feature maps connected by three pieces of optical flow information after MOFRW blocks, as shown in Figure 6. We used Grad-CAM [41] as our visualization method, and by looking at the color distribution in the heat map, we can find the correlation, importance, and degree of influence between different features. Visualizing the heat map helps us better understand the model’s prediction results and the impact of each feature on the prediction results. The validity of the model was verified in both sets of images by comparing the detailed information of the heat maps with the original optical flow images. For emotions of depression, surprise, and disgust, which have little difference in muscle motor sites, the model achieved precise localization of the motor sites. For the emotion of happiness, which has significant differences in muscle motor sites (such as eyebrows and mouth), the model still accurately focused on the motor regions. These visualized heat maps further enhance our confidence in the model.

## 5. Conclusions

In this study, a model called AM3F-FlowNet is proposed, which can effectively solve the problem of difficulty in extracting subtle local features in micro-expression recognition tasks. The model combines multi-scale, multi-channel, and attention mechanism techniques to exploit the complementary characteristics of different optical flows and can thoroughly learn the motion information of micro-expressions and extract the salient and distinguishing features of micro-expressions. We designed several modules, specifically the MSFF and MOFRW modules. Among them, the MSFF module specializes in highlighting subtle local movements of the face. In contrast, the MOFRW module uses the contribution of different optical flows for reweighting to extract key features and filter redundant features. By visualizing the high-level feature maps, we demonstrated the effectiveness of the attention module in capturing facial regions associated with emotions. During the experiments, we introduced a logit adjustment loss function to successfully mitigate the category imbalance problem of the micro-expression dataset. The experimental results show that the performance of the AM3F-FlowNet model on multiple micro-expression datasets is comparable to that of state-of-the-art methods, which fully demonstrates the effectiveness of the algorithm proposed in this paper. In the future, we will further investigate the semantic correlation of local micro-expression features to enhance the model’s interpretability and feature-learning capability.

## Figures and Tables

**Figure 1 entropy-25-01064-f001:**
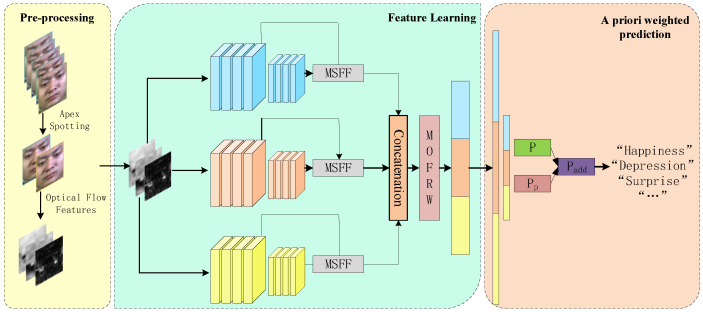
The general flow chart of micro-expression recognition. The model is divided into three parts: data preprocessing, feature learning, and a priori weighted prediction. Among them, the data preprocessing part contains apex frame localization and optical flow feature calculation, and the feature learning part uses inception blocks for feature extraction, with 6 convolutional kernels for each scale in the first Inception block and 16 convolutional kernels for each scale in the second inception block. In addition, the multi-scale feature fusion (MSFF) and multi-optical flow feature reweighting (MOFRW) modules are used to fuse features and reweight optical flow features, respectively. In the prior weighting part, the model uses the prior probability ($P_{p}$) learned from the dataset distribution and adds it to the model prediction to obtain the final prediction ($p_{add}$).

**Figure 2 entropy-25-01064-f002:**
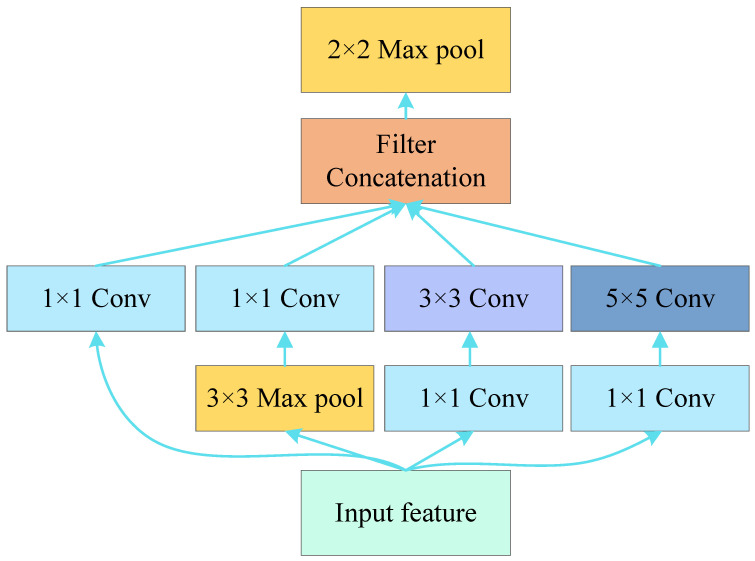
The structure of the inception block.

**Figure 3 entropy-25-01064-f003:**
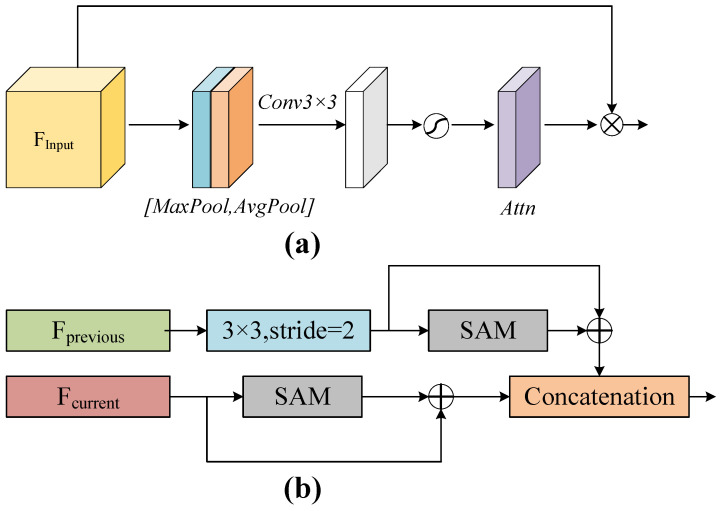
(**a**) SAM flow chart; (**b**) MSFF flow chart.

**Figure 4 entropy-25-01064-f004:**
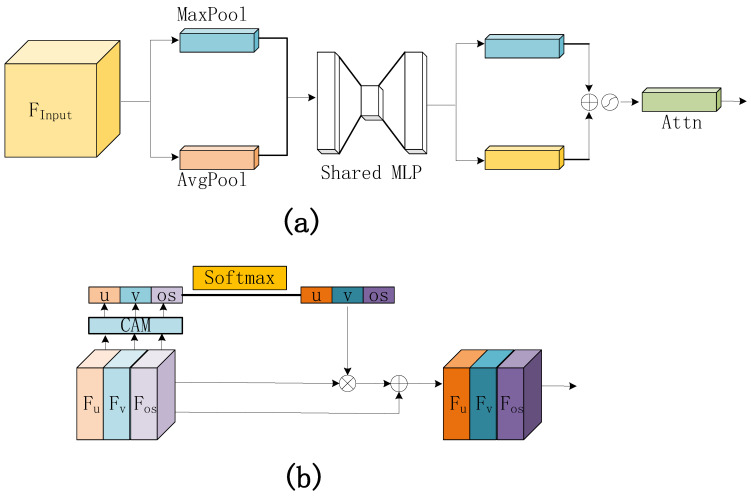
(**a**) CAM flow chart; (**b**) MOFRW flow chart.

**Figure 5 entropy-25-01064-f005:**
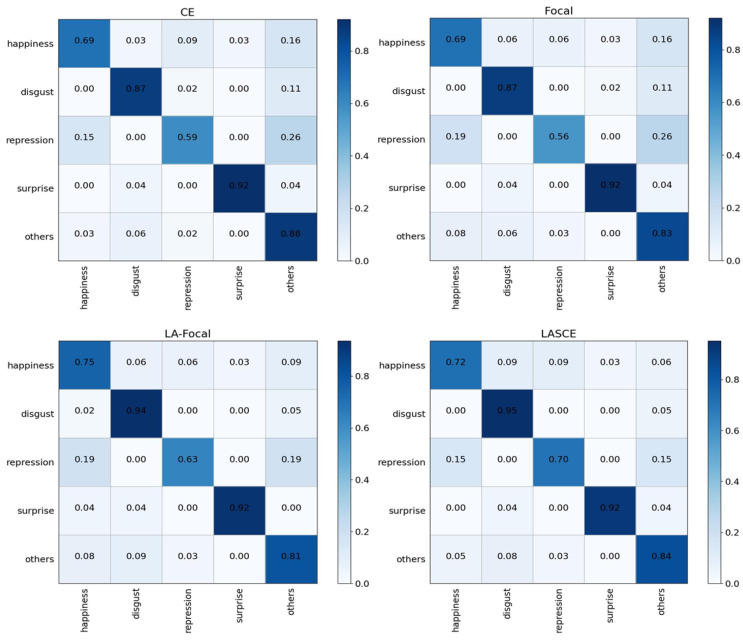
Confusion matrix for classification results of different loss functions. For the CASMEII dataset, the number of disgust categories is 60, the number of happiness categories is 32, the number of repression categories is 27, the number of surprise categories is 25, and the number of other categories is 102.

**Figure 6 entropy-25-01064-f006:**
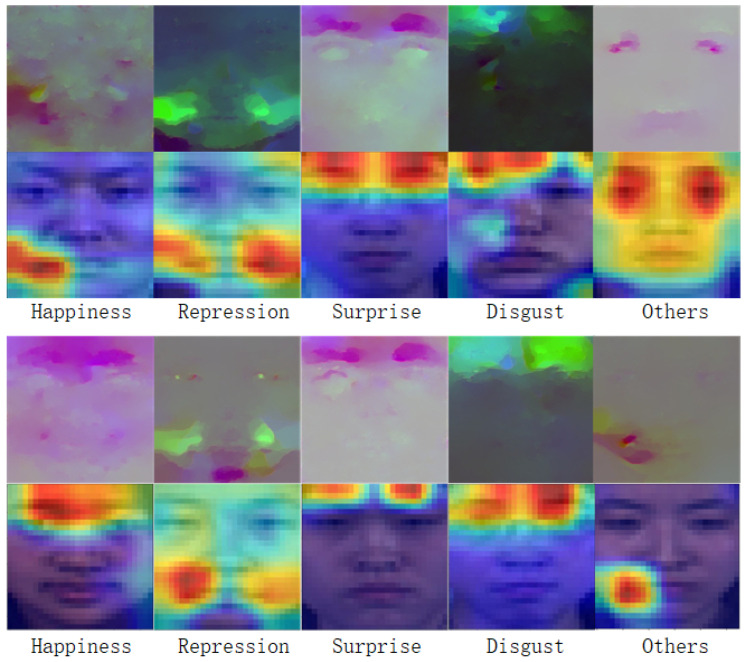
Raw optical-flow–optical-strain image with the visualized heat map.

**Table 1 entropy-25-01064-t001:** The specific information of the combined dataset.

Dataset		SAMM	SMIC-HS	CASMEII	3DB-Combined
Number of subjects		28	16	24	68
Number of samples		133	164	145	442
Expression	Positive	26	51	32	109
	Negtive	92	70	88	250
	Surprise	15	43	25	83

**Table 2 entropy-25-01064-t002:** Performance comparison on a single dataset.

Method	SAMM (5 Classes)		SMIC-HS (3 Classes)		CASMEII (5 Classes)	
	*Acc*	*F*1	*Acc*	*F*1	*Acc*	*F*1
LBP-TOP [8]	-	-	53.66	0.5384	46.46	0.4241
MDMO [12] (2016)	-	-	61.5	0.406	51.0	0.418
Bi-WOOF [13] (2018)	59.8	0.591	59.3	0.620	58.9	0.610
DSSN [31] (2019)	57.35	0.4644	63.41	0.6462	70.78	0.7297
STRCN [32] (2019)	54.5	0.492	53.1	0.514	56.0	0.542
SLSTT [33] (2021)	**72.388**	**0.640**	73.17	0.724	75.806	0.753
GEME [34] (2021)	55.88	0.4538	64.63	0.6158	75.2	0.7354
Later [35] (2022)	-	-	73.17	**0.7447**	70.68	0.7106
FDCN [36] (2022)	58.07	0.57	-	-	73.09	0.72
KTGSL [37] (2022)	-	-	72.58	0.6820	75.64	0.6917
AM3F-FlowNet (Ours)	66.18	0.5410	**74.25**	0.7254	**84.52**	**0.8288**

**Table 3 entropy-25-01064-t003:** Performance comparison on a combined dataset.

Method	Full		SAMM		SMIC		CASMEII	
	*UF*1	*UAR*	*UF*1	*UAR*	*UF*1	*UAR*	*UF*1	*UAR*
LBP-TOP [8]	0.5882	0.5280	0.3954	0.4102	0.2000	0.5280	0.7026	0.7429
Bi-WOOF [13] (2018)	0.6296	0.6227	0.5211	0.5139	0.5727	0.5829	0.7805	0.8026
OFF-ApexNet [16] (2019)	0.7196	0.7096	0.5409	0.5392	0.6817	0.6695	0.8764	0.8681
Dual-Inception [18] (2019)	0.7322	0.7278	0.5868	0.5663	0.6645	0.6726	0.8621	0.8560
CapsuleNet [15] (2019)	0.6520	0.6506	0.6209	0.5989	0.5820	0.5877	0.7068	0.7018
STSTNet [17] (2019)	0.7353	0.7605	0.6588	0.6810	0.6801	0.7013	0.8382	0.8686
EMR [30] (2019)	0.7885	0.7824	0.7754	0.7152	0.7461	0.7530	0.8293	0.8209
STA-GCN [38] (2021)	-	-	-	-	-	-	0.7608	0.7096
SLSTT [34] (2021)	0.8160	0.7900	0.7150	0.6420	0.7240	0.7070	0.9010	0.8850
AUGCN [39] (2021)	0.7914	0.7933	0.7392	0.7163	0.7651	0.7780	0.9071	0.8878
BDCNN [40] (2022)	0.8509	0.8500	**0.8538**	**0.8507**	0.7859	0.7869	0.9501	0.9516
AM3FFlowNet (Ours)	**0.8536**	**0.8594**	0.7643	0.7452	**0.7946**	**0.7941**	**0.9591**	**0.9567**

**Table 4 entropy-25-01064-t004:** Accuracy results of AM3F-FlowNet and its variants (AM3F-FlowNet without multi-scale, AM3F-FlowNet using multi-scale, and AM3F-FlowNet using MSFF blocks) on the CASMEII dataset. For each classification task, the highest performance is highlighted in bold.

Method	CASMEII (5 Classes)	
	*Acc* (%)	*F*1
AM3F-FlowNet without multi-scale	81.17	0.7905
AM3F-FlowNet without MSFF	83.26	0.8231
AM3F-FlowNet without MOFRW	81.59	0.8092
AM3F-FlowNet	**84.52**	**0.8288**

**Table 5 entropy-25-01064-t005:** Accuracy results of AM3F-FlowNet on the CASMEII dataset using each of the four loss functions. For each classification task, the highest performance is highlighted in bold.

Loss	CASMEII (5 Classes)	
	*Acc* (%)	*F*1
CE	82.38	0.8050
Focal	79.92	0.7799
LA-Focal	82.43	0.8048
LASCE	**84.52**	**0.8288**

## Data Availability

Not applicable.

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
