# Peer review of "AM3F-FlowNet: Attention-Based Multi-Scale Multi-Branch Flow Network"

_entropy, 2023, doi:10.3390/e25071064_

Round 1

Reviewer 1 Report

The paper is on the topic multiscale and multibranch performance boosting. 

In Table 2, for CASMEII(5classes) the proposed method's performance is too high compared to other methods. ~10% boost. Why there is not such significant improvements in other two datasets? Is it due to the dataset level or is it due to class size. Justifications are missing. 

Results on combination of datasets are not boosting the performance too drastically. Is it due to? 

Tables 4-5 are not very clear what author want to show. 

As an observation, MOFRW contributes majorly. And MSFF is the least. I believe these tables can be combined for better understanding. 

Table 6 is about loss functions. And shows that LASCE outperforms compared to other. WHy? 

Ablations studies can be scaled up to more experiments? 

We need more visualizations to make a conclusion. 

References are adequate but can be further updated. 

Minor grammatical corrections needed. 

Reviewer 2 Report

The paper proposes a deep-learning architecture to integrate three kinds of optical flow information (horizontal optical flow, vertical optical flow, and optical strain) for recognizing facial micro-expressions. While the paper present a rich set of results, I have the following questions/suggestions.

[Major]

1. Can the authors provide direct evidence of the effectiveness of the proposed LASCE, namely mitigation of the class imbalance problem? (e.g., showing some confusion matrices)

2. Do Tables 2 & 3 present the max, mean, or one-time performance for each method on each dataset? In Table 3, the proposed AM3FFlowNet does not seem to outperform BDCNN in statistical sense. So the authors may want to run some statistical tests and discuss whether AM3FFlowNet outperforms BDCNN in terms of accuracy/F1 or efficiency.  

3. Do the authors cherry-pick example images for Fig.5? How consistent are the heat maps from Grad-CAM on different images of the same category?

[Minor]

4. The authors may want to discuss/cite this review article:

Ben, X., Ren, Y., Zhang, J., Wang, S. J., Kpalma, K., Meng, W., & Liu, Y. J. (2021). Video-based facial micro-expression analysis: A survey of datasets, features and algorithms. IEEE transactions on Pattern Analysis and Machine Intelligence, 44(9), 5826-5846.

Round 2

Reviewer 1 Report

The paper is not in a good shape. Author has responded to all raised concerns and I have not further questions. 

Reviewer 2 Report

The authors have addressed all my concerns.